# Structure and transport mechanism of the sodium/proton antiporter MjNhaP1

**Cristina Paulino[†], David Wöhlert[†], Ekaterina Kapotova, Özkan Yildiz\*, Werner Kühlbrandt\***

Department of Structural Biology, Max Planck Institute of Biophysics, Frankfurt am Main, Germany

**Abstract** Sodium/proton antiporters are essential for sodium and pH homeostasis and play a major role in human health and disease. We determined the structures of the archaeal sodium/proton antiporter MjNhaP1 in two complementary states. The inward-open state was obtained by x-ray crystallography in the presence of sodium at pH 8, where the transporter is highly active. The outward-open state was obtained by electron crystallography without sodium at pH 4, where MjNhaP1 is inactive. Comparison of both structures reveals a 7° tilt of the 6 helix bundle. $^{22}Na^+$ uptake measurements indicate non-cooperative transport with an activity maximum at pH 7.5. We conclude that binding of a $Na^+$ ion from the outside induces helix movements that close the extracellular cavity, open the cytoplasmic funnel, and result in a ~5 Å vertical relocation of the ion binding site to release the substrate ion into the cytoplasm.

## Introduction

Na$^+$/H$^+$ antiporters are essential secondary-active transporters of the cation-proton antiporter (CPA) family (*Brett et al., 2005*). CPA antiporters are conserved across all biological kingdoms and play crucial roles in pH, ion and volume homeostasis (*Padan, 2013*). The CPA1 branch of the family includes the archaeal NhaP antiporters from *Methanocaldococcus jannaschii* (MjNhaP1) and *Pyrococcus abyssii* (PaNhaP) and the medically important human NHE sodium proton exchangers (*Donowitz et al., 2013*; *Fuster and Alexander, 2014*). CPA1 antiporters are electroneutral and exchange one Na$^+$ against one H$^+$ (*Calinescu et al., 2014*; *Wöhlert et al., 2014*). CPA2 antiporters, including EcNhaA from *E. coli* and TtNapA from *Thermus thermophilus* are electrogenic, exchanging one Na$^+$ against two H$^+$ (*Lee et al., 2013a*; *Taglicht et al., 1993*). Previous electron crystallographic studies have shown the structure of the MjNhaP1 dimer in the membrane at 7 Å resolution (*Vinothkumar et al., 2005*; *Goswami et al., 2011*) and revealed substrate-induced conformational changes within the range of physiological Na$^+$ concentrations and pH. MjNhaP1 shares significant sequence homology of functionally important regions with the mammalian NHEs (*Goswami et al., 2011*). MjNhaP1 and NHE1 are both thought to use a sodium gradient to maintain the intracellular pH by expelling protons from the cell (*Lee et al., 2013b*; *Paulino and Kühlbrandt, 2014*), but the mechanism by which this happens has remained unknown.

## Results

### X-ray structure of MjNhaP1

MjNhaP1 has 13 transmembrane helices (TMH), referred to as H1-13. The N-terminal H1 is essential for transport activity (*Goswami et al., 2011*), but its orientation in the membrane has not been determined experimentally. Comparison to the EcNhaA structure predicts the cytoplasmic location of the MjNhaP1 C-terminus. We performed GFP/PhoA activity assays with MjNhaP1 expressed in *E. coli*, which indicated that the C-terminus is indeed on the cytoplasmic side (*Figure 1*). Because there are 13 membrane spans, the N-terminus is on the extracellular side.

**\*For correspondence:** Oezkan.
Yildiz@biophys.mpg.de (ÖY);
werner.kuehlbrandt@biophys.
mpg.de (WK)

[†]These authors contributed equally to this work

**eLife digest** Although the membrane that surrounds a cell is effective at separating the inside of a cell from the outside environment, certain molecules and ions must enter or leave the cell for it to work correctly. Proteins embedded in the cell membrane, called transporters, ensure this occurs.

Transporters that are found in all organisms include the sodium/proton antiporters, which exchange protons from inside the cell with sodium ions from outside. However, exactly how these antiporters work was unknown.

Paulino, Wöhlert et al. have now examined the structure of a sodium/proton antiporter from a single-celled organism called *Methanocaldococcus jannaschii*, a species of archaea that thrives at high temperature. Using X-ray crystallography, Paulino, Wöhlert et al. uncovered the structure of the antiporter in the presence of sodium ions and in alkaline conditions. Under these conditions the sodium/proton antiporter adopts an 'inward-open' state, where the substrate-binding site—the region where the ions bind to be transported—of the transporter is open towards the cell interior. Paulino, Wöhlert et al. also used electron cryo-microscopy to investigate the antiporter's structure under acidic conditions in the absence of salt. This revealed an 'outward-open' state, where the substrate-binding site of the transporter is open towards the space outside of the cell. The main difference between this and the inward-open state is the movement of a bundle of six helices within the antiporter. This activating structural change occurs when a sodium ion binds to the antiporter rather than by a change in acidity.

Paulino, Wöhlert et al. found that the structure of the *M. jannaschii* antiporter is very similar to the structures of an antiporter from another archaea species, which was studied in separate work. The acidity range under which the two transporters are most active is different, indicating that minor changes in the amino acid sequence that make up their structure can have a substantial effect on the activity of these antiporters.

The next step will be to use computer simulations to calculate how sodium/proton antiporters change from an inward-open to an outward-open state. The *M. jannaschii* antiporter will be particularly suitable for such simulations, as Paulino, Wöhlert et al. found that it transports ions more rapidly than any previously known transporter. Understanding how these transporters work is also medically relevant, as defects in related sodium/proton antiporters in humans are implicated in serious and life-threatening diseases.

The structure of MjNhaP1 was determined by molecular replacement with the related PaNhaP (*Wöhlert et al., 2014*) (pdb 4cz8), using crystals grown with 100 mM NaCl at pH 8. The two dimers in the asymmetric unit were refined to 3.5 Å resolution (*Table 1*). The 13 TMHs in the monomer are arranged into a 6-helix bundle and a row of seven helices at the dimer interface, as in PaNhaP, but the two structures differ in important details. Seen from above or below the membrane, the MjNhaP1 dimer is oval (*Figure 2A*, *Figure 2—figure supplement 1A*) rather than rectangular. Seen from the side, it is about 10 Å shorter than PaNhaP (*Figure 2B*, *Figure 2—figure supplement 1B*). Its cytoplasmic surface is flat and does not extend more than 10 Å above the membrane. The cytoplasmic cavity protrudes only 13 Å into the interior of the protomer. On the extracellular side, a negatively charged funnel, which is considerably wider and deeper than in PaNhaP, extends 15 Å towards the center of the protomer (*Figure 2—figure supplement 2*). A short loop on the extracellular side of MjNhaP1 connects H6 to H7 and a short amphipathic helix connects H12 to H13 above the center of the helix bundle, whereas in PaNhaP, H6 and H7 are connected by a helix and H12 and H13 by a short loop (*Video 1*).

## Dimer interface

While the 6-helix bundle of MjNhaP1 closely resembles that of PaNhaP, there are significant differences at the dimer interface. H10 of MjNhaP1 is much shorter and does not protrude into the cytoplasm. MjNhaP1 has no equivalent for His292 in H10, a key residue for transport in PaNhaP (*Wöhlert et al., 2014*). On the cytoplasmic side of the MjNhaP1 dimer interface, a line of ionic and polar residues parallel to the membrane surface maintains tight interactions between protomers. On the extracellular side, dimer interactions are mediated largely by the hydrophobic H1, although deletion of this helix, which is necessary for function, did not abrogate dimer formation (*Goswami et al., 2011*).

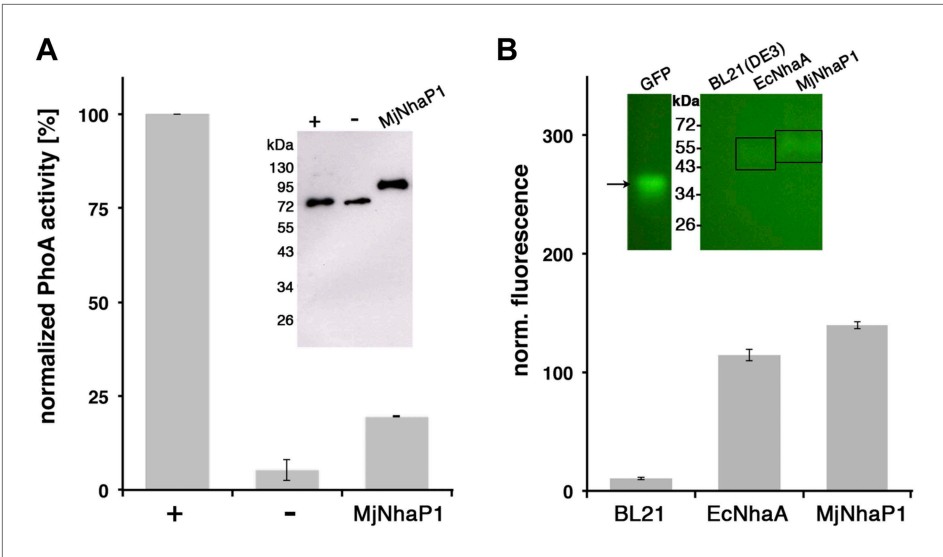

**Figure 1**. Membrane topology of MjNhaP1. (**A**) PhoA activity assay. Enzymatic activity of a C-terminal MjNhaP1 fusion construct with alkaline phosphatase expressed in the PhoA-deficient *E. coli* strain CC118 was measured spectroscopically. PhoA is active only in the periplasm. (+) PhoA fused to the periplasmic C-terminus of YiaD or to the cytoplasmic C-terminus of YedZ were used as positive (+) or negative (−) controls. Western blot analysis (inset) shows that constructs were expressed at comparable levels. (**B**) GFP assay. Normalized whole-cell fluorescence of C-terminal MjNhaP1-GFP-fusion constructs expressed in BL21(DE3) cells. GFP is fluorescent only in the cytoplasm. EcNhaA, which has a cytoplasmic C-terminus, was used as positive control and untransformed BL21(DE3) cells as negative control. In-gel florescence of the constructs is shown in the inset. The activity of the positive control was set to 100%. The low activity of the PhoA construct, together with the fluorescence of the GFP fusion construct indicates that the C-terminus of MjNhaP1 expressed in E. coli is on the cytoplasmic side.

Mutational analysis indicates that the first 15 N-terminal residues of H1 are dispensable for activity (*Figure 2—figure supplement 3*). On the dimer interface of MjNhaP1 there is a deep, hydrophobic cavity, which, unlike that in PaNhaP, spans nearly the entire thickness of the membrane and is covered on the extracellular side (*Figure 2—figure supplement 2*). As in PaNhaP, this cavity contains density indicative of co-purified, flexible lipids, which was however not sufficiently well-defined for building an atomic model.

## Substrate binding and transport

Structural homology to PaNhaP indicates that the substrate ion, which is not resolved in MjNhaP1, is coordinated by Asp132 and the backbone of Thr131 in the unwound stretch of H5. Other key residues in the ion-binding site are Ser157 and Asp161 in H6 (Ser155 and Asp159 in PaNhaP; *Figure 3*). MjNhaP1 lacks an equivalent of the ion-coordinating Glu73 in H3 of PaNhaP, which is however not essential for transport (*Wöhlert et al., 2014*). Instead, Thr76 in H3 of MjNhaP1 interacts with Glu154 in H6, which may guide the substrate ion from the binding site to the cytoplasm. Asn160 in H6 is part of the characteristic ND motif in the CPA1 antiporters (*Figure 2—figure supplement 4*, *Figure 3—figure supplement 1*, *Figure 3—figure supplement 2*). Comparison with PaNhaP suggests that Asn160 is unlikely to coordinate the substrate ion directly. Rather, it forms a hydrogen bond to the hydroxyl group of Thr131, which in PaNhaP does participate in substrate-ion coordination via its main-chain carbonyl. Nevertheless, changing N160 to alanine renders MjNhaP1 inactive (*Figure 4A*). A mutant in which N160 is exchanged against aspartate has reduced activity but is not electrogenic (*Figure 4B,C*).

Measurements of $^{22}$Na$^+$ uptake by wildtype MjNhaP1 reconstituted into proteoliposomes indicate an activity maximum at pH 7.5 (*Figure 5A*). Transport increases by a factor of approximately two from 0.94 ions per second per protomer at pH 6 (*Figure 5B*) to 1.68 ions per second at pH 8 (*Figure 5C*) at a $K_m^{Na+}$ of 0.84 mM. Unlike PaNhaP, MjNhaP1 is not cooperative at any pH tested. Linear plots (*Figure 5—figure supplement 1*) show that near-saturation is reached around 5 mM NaCl at either pH.

**Table 1.** X-ray crystallographic data

|  | Native |
| --- | --- |
| Data collection |  |
| Wavelength | 0.976 |
| Space group | P2₁ |
| Cell dimensions |  |
| $a, b, c$ (Å) | 98.4, 102.5, 132.1 |
| α, β, γ (°) | 90.0, 105.6, 90.0 |
| Resolution (Å) | 31.93–3.5 (3.72–3.5) |
| $R_{pim}$ | 0.086 (0.573) |
| $I / \sigma I$ | 6.0 (1.7) |
| CC* | 0.999 (0.918) |
| Completeness (%) | 99.3 (99.2) |
| Multiplicity | 7.0 (7.2) |
| Refinement |  |
| Resolution (Å) | 31.93–3.5 (3.72–3.5) |
| Unique reflections | 58,249 |
| Reflections in test set | 3115 |
| $R_{work}/R_{free}$ (%) | 25.2/30.2 (34.4/39.2) |
| CC(work)/CC(free) | 0.905/0.930 (0.799/0.645) |
| Wilson B-Factor (Å²) | 141 |
| Atoms in asymmetric unit | 12,548 |
| Protein | 12,535 |
| Ligands | 13 |
| r.m.s. deviations: |  |
| Bond lengths (Å) | 0.003 |
| Bond angles (°) | 0.785 |

## 3D EM structure of MjNhaP1

3D crystals of MjNhaP1 without sodium or at low pH either failed to grow or were poorly ordered. We therefore determined the structure of the sodium-free state at low pH by electron cryo-crystallography of 2D crystals grown at pH 4 without NaCl (*Paulino and Kühlbrandt, 2014*). Amplitudes and phases obtained from 128 images were merged to yield a 3D map (*Table 2*, *Figure 6*) with an in-plane resolution of 6 Å (*Figure 6—figure supplement 1*). The unit cell contained two dimers, with one protomer in the asymmetric unit. The MjNhaP1 x-ray structure was fitted manually to the EM map to obtain an atomic model (*Figure 7*). A 3D difference map calculated between the experimental 6 Å EM density and the x-ray map truncated to this resolution indicated clear changes in the orientation of key helices (*Figure 8*). Of the interface helices, only H10 required a tilt of ~6° about the helix center. Within the 6-helix bundle, H6 was tilted by ~14° and H12$_E$ by ~15°. H5$_C$ and H5$_E$ changed direction by ~15° and ~7°, respectively. The 6-helix bundle as a whole tilted by ~7° towards the dimer interface on the cytoplasmic side and away from the dimer interface on the extracellular side, about an axis in the membrane plane roughly parallel to the dimer interface. On the cytoplasmic side of the EM model, H5$_C$ and H6 are closer to the interface than in the x-ray structure and obstruct access to the substrate-binding site. On the extracellular side, the tilt of the 6-helix bundle, especially of H6 and 12$_E$, widens and deepens the exterior funnel. Whereas the x-ray structures of MjNhaP1 and PaNhaP (*Wöhlert et al., 2014*) both show the inward-open conformation, the EM structure of MjNhaP1 is closed on the cytoplasmic and open on the extracellular side (*Figure 9*). We refer to this conformation as an 'outward-open' state of MjNhaP1. In the transition from inward-open to outward-open, the ion-binding site moves towards the extracellular side by about 5 Å (*Figure 9*).

## Discussion

### Inward-open and outward-open states

A projection difference map between the x-ray structure and the EM model calculated at 6 Å resolution (*Figure 10A–C*) indicates significant lateral changes in helix position and orientation in the 6-helix bundle, whereas the dimer interface changes only minimally. It is instructive to compare this difference map to Figure 5 of an earlier paper that describes substrate-ion induced conformational changes in MjNhaP1 (*Paulino and Kühlbrandt, 2014*). *Figure 10* shows that the positions and relative strength of difference peaks between the inward-open and outward-open state are nearly identical to those that are observed when the 2D crystals of MjNhaP1 are taken from 0 mM NaCl to 500 mM NaCl, either at pH 8 or at pH 4. Together with the 3D data presented here, this allows us to conclude that an increase in NaCl concentration converts the antiporter from the outward-open conformation in the absence of salt to the inward-open conformation in the presence of salt.

Comparison of the MjNhaP1 inward-open x-ray structure to the 3.45 Å x-ray structure of EcNhaA reveals similarities in the 6-helix bundle (*Goswami et al., 2011*), but clear differences at the dimer interface and its position relative to the 6-helix bundle. The outward-open state of the MjNhaP1 EM

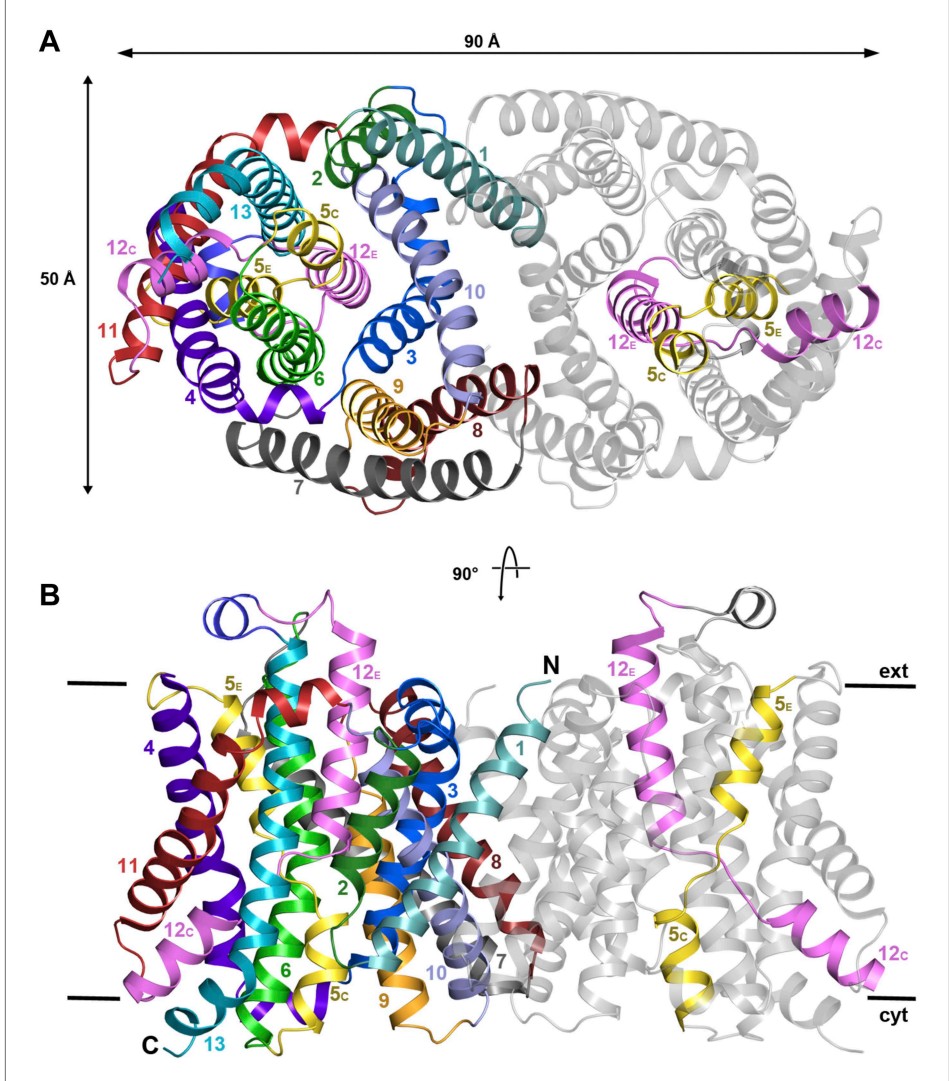

**Figure 2**. X-ray structure of MjNhaP1. (**A**) The MjNhaP1 dimer seen from the cytoplasmic side. Helices H1 to 13 are color-coded and numbered. In one protomer, only the partly unwound helices H5 and 12 are colored. (**B**) Side view with the N-terminus of H1 on the extracellular side.

The following figure supplements are available for figure 2:

**Figure supplement 1**. X-ray structure of MjNhaP1.

**Figure supplement 2**. Cavities in MjNhaP1 at pH 8.

**Figure supplement 3**. Na$^+$/H$^+$ antiport activity of H1 truncation mutants.

**Figure supplement 4**. Sequence alignment of CPA antiporters.

structure is confirmed by comparison with the outward-open structure of the CPA2 antiporter TtNapA (*Lee et al., 2013a*), which looks strikingly similar, especially with respect to the extracellular funnel (*Figure 9—figure supplement 1A*), the conformation of H6 and the H5/12 pairs of half helices (*Figure 9—figure supplement 1B*).

The structure of the apical sodium-dependent bile acid symporter ASBT, which surprisingly has the same fold as the sodium/proton antiporters, has been solved in both the inward-open and the outward-open state (*Hu et al., 2011*; *Zhou et al., 2014*). Comparison reinforces our conclusion that

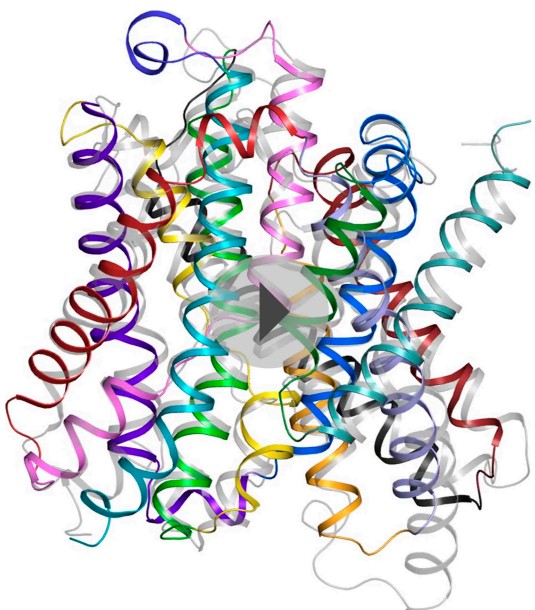

**Video 1**. Comparison of the x-ray structures of MjNhaP1 and PaNhaP. One protomer of MjNhaP1 (colored) is superposed on one protomer of PaNhaP (grey). DOI: 10.7554/eLife.03583.010

the x-ray structure of MjNhaP1 shows the inward-open and the EM structure the outward-open state (*Figure 9*). In both MjNhaP1 and ASBT, the 6-helix bundle performs a rigid-body rotation around the same axis in the membrane parallel to the dimer interface (*Video 2*). The resulting up-and-down movement of the substrate-binding site is more pronounced in ASBT than in MjNhaP1, as might be necessary to facilitate translocation of the larger bile acid substrate.

A different and much larger conformational change has been postulated for TtNapA on the basis of its outward-open x-ray structure and an inward-open state modeled on the dissimilar EcNhaA structure (*Lee et al., 2013a*). The inward-open model of TtNapA implied that the 6-helix bundle moves up and down by 10 Å and rotates by 21° about an axis roughly perpendicular to that in MjNhaP1 and ASBT. The similarity of the MjNhaP1 EM structure to the TtNapA x-ray structure suggests however that the inward-open state of TtNapA closely resembles the x-ray structure of MjNhaP1 rather than that of EcNhaA. We conclude that all sodium/proton antiporters undergo essentially the same conformational changes in the course of their transport cycles, as represented here by the two states of MjNhaP1.

## Structural differences between electrogenic and electroneutral antiporters

The electroneutral CPA1 and electrogenic CPA2 antiporters have different ion transport stoichiometries. CPA2 antiporters, such as EcNhaA and TtNapA, exchange two protons against one Na$^+$. One of the predicted motifs for the electrogenic transport is the DD motif in helix V in place of the ND motif in H6 of CPA1 antiporters, such as MjNhaP1 and PaNhaP (*Figure 2—figure supplement 4*, *Figure 3—figure supplement 1*, *Figure 3—figure supplement 2*). In CPA2 antiporters the two conserved aspartates have been proposed to each bind one of the translocated protons (*Taglicht et al., 1991*; *Hunte et al., 2005*; *Arkin et al., 2007*). However, replacing N160 in the ND motif does not render MjNhaP1 electrogenic (*Figure 4C*). The inactive N160A mutant (*Figure 4A*) shows that this sidechain is important for ion translocation, even though it does not participate in ion coordination directly. The reduced activity of the N160D mutant (*Figure 4B,C*) suggests a possible role in stabilizing the proton or substrate-bound state, which can also be fulfilled by an aspartate. Note that an asparagine in this position renders EcNhaA and TtNapA inactive (*Inoue et al., 1995*; *Lee et al., 2013a*), probably because it cannot form an ion bridge, as observed for Lys305 and Asp156 in TtNapA (*Lee et al., 2013a*), which may be important for protein stability (*Figure 3—figure supplement 2*). In MjNhaP1 and PaNhaP the arginine replacing this lysine does not interact with the ND motif but forms an ion bridge to the neighboring conserved glutamate in H6 (*Figure 3A* and *Figure 3—figure supplement 1*), which would stabilize the 6-helix bundle. In terms of its overall structure, TtNapA is more similar to MjNhaP1 and PaNhaP than to EcNhaA (*Hunte et al., 2005*), especially with respect to the relative position of the dimer interface with the seven helices. The tertiary structure of CPA antiporters thus does not correlate with their transport stoichiometry. There appear to be two types, one of which, represented by MjNhaP1, PaNhaP and TtNapA, is more common than the other type, that seems to be confined to EcNhaA and its close relatives.

## Transport rates

$^{22}$Na$^+$ uptake measurements with MjNhaP1 proteoliposomes acidified by an ammonium sulfate gradient indicated a bell-shaped pH profile with a pH maximum at around pH 7.5. Activity dropped to

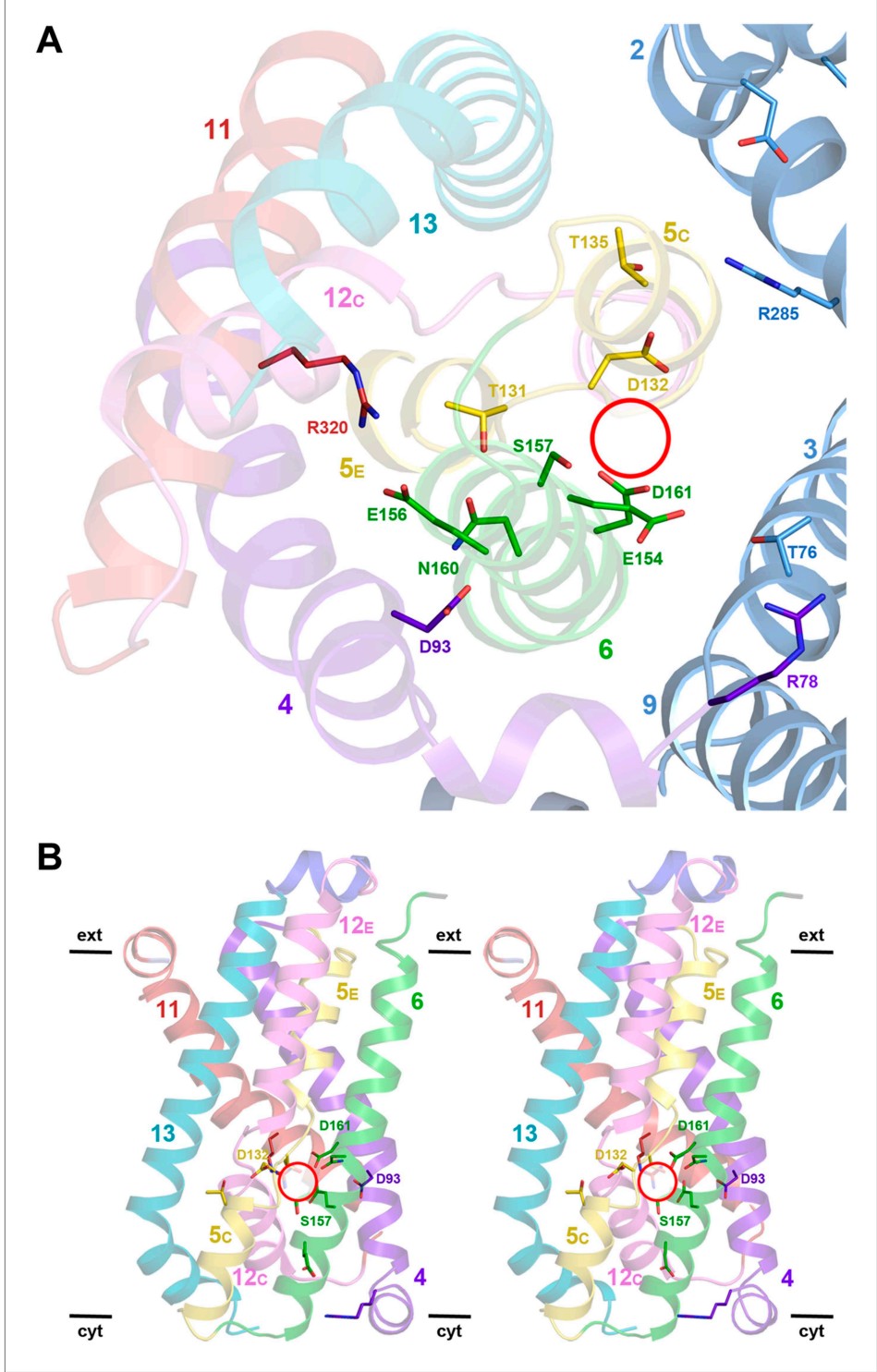

**Figure 3**. Detailed views of the MjNhaP1 x-ray structure. (**A**) Cytoplasmic view of the 6-helix bundle with residues involved in substrate-ion binding and transport. The essential N160 in the ND motif points away from the ion-binding site and interacts with Asp93. Glu154 in H6 interacts with Thr76 in the interface helix H3. Thr76 replaces the ion-coordinating Glu73 of PaNhaP. The 6-helix bundle interacts via Thr135 in H5$_C$ with Arg285 in H10 and Asp31 in H2. Arg320 forms an ion bridge with Glu156. (**B**) Stereo view of the 6-helix bundle. The red circle marks the ion-binding site in PaNhaP (*Wöhlert et al., 2014*).

*Figure 3. Continued on next page*

*Figure 3. Continued*

The following figure supplements are available for figure 3:

**Figure supplement 1**. 6-helix bundle in the CPA1 antiporters MjNhaP1 and PaNhaP.

**Figure supplement 2**. 6-helix bundle in the CPA2 antiporters EcNhaA and TtNapA.

background levels below pH 4 or above pH 9. Earlier studies (*Hellmer et al., 2003*; *Vinothkumar et al., 2005*; *Goswami et al., 2011*) had found that MjNhaP1 is active at pH 6 but inactive at pH 7.5 or above. This discrepancy is due to the C-terminal affinity tag on the construct that was used in previous transport measurements (*Hellmer et al., 2003*; *Goswami et al., 2011*). We repeated the measurements with this tagged construct under symmetrical pH conditions and found that it was indeed inactive at pH 8 but active at pH 6 (*Figure 4D*). Apparently the affinity tag at the C-terminus of H13,

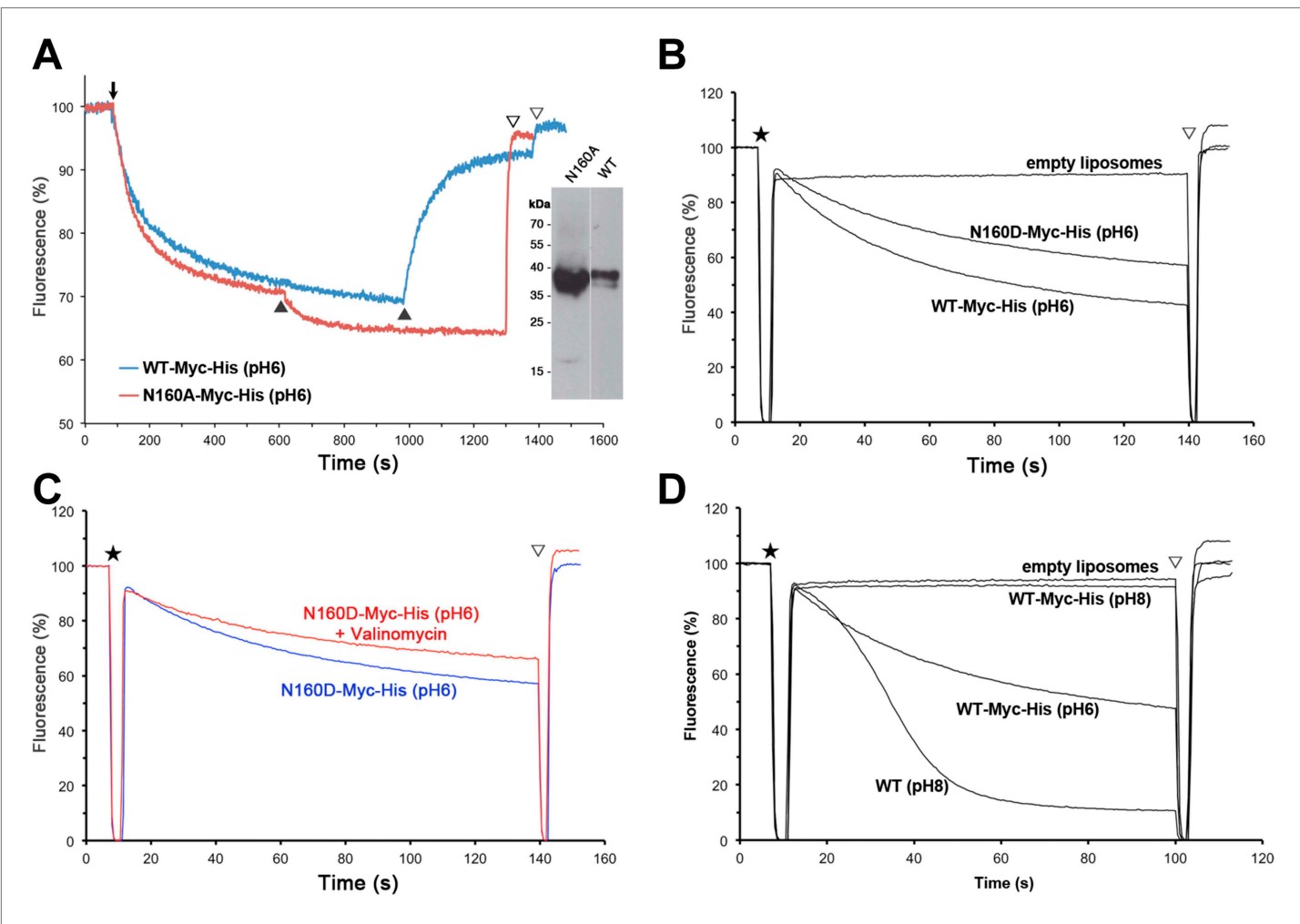

**Figure 4**. Activity of Asn160 mutants. (**A**) Na$^+$/H$^+$ antiporter activity of wildtype (WT) and the N160A mutant measured under asymmetrical conditions in everted vesicles at pH 6. Constructs were expressed in Na$^+$/H$^+$ antiporter-deficient KNabc cells at comparable levels (inset). Everted vesicles were preloaded with protons by addition of 2.5 mM Tris/DL-lactate (↓). Transport was initiated by addition of NaCl to 25 mM (▲), and proton efflux was monitored by acridine orange fluorescence dequenching. (**B**) Activity of purified wildtype MjNhaP1 (WT) and the N160D mutant reconstituted into proteoliposomes under symmetrical pH. (**C**) Activity at pH 6 was unaltered in the presence of valinomycin, demonstrating that N160D is not electrogenic. Asterisks in (**B**) and (**C**) mark the addition of proteoliposomes. The pH gradient was dissipated with 25 mM NH4Cl (▽). (**D**) Transport by MjNhaP1 is inhibited by the Myc-His-Tag at pH 8 but not at pH 6.

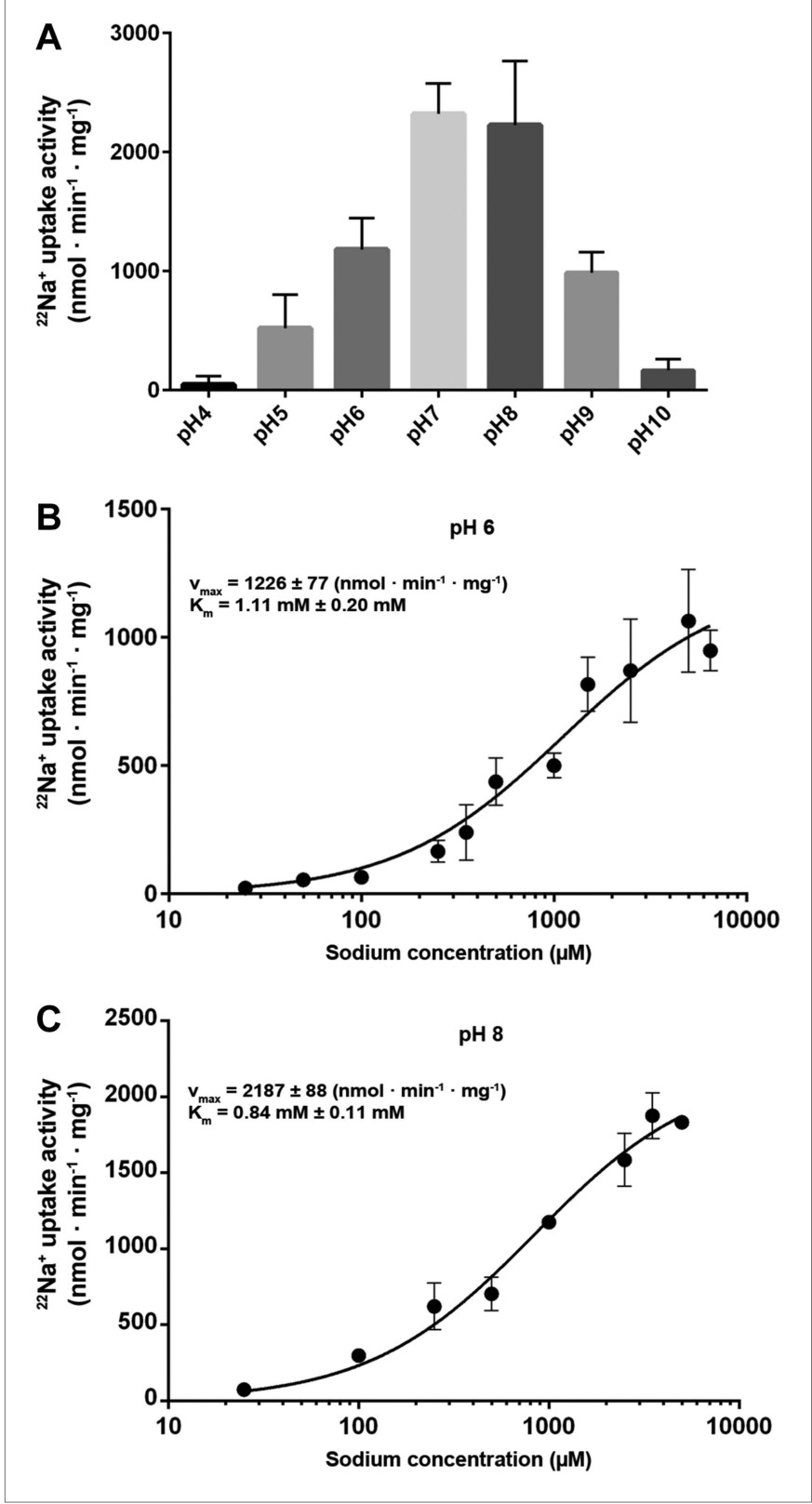

**Figure 5**. Transport activity of MjNhaP1. (**A**) pH profile measured by $^{22}$Na uptake with acidic-inside proteoliposomes at 1.5 mM NaCl. MjNhaP1 activity is highest between pH 7 and pH 8 and drops to background level at pH 4 and pH 10. (**B**) $^{22}$Na$^+$ transport at pH 6 follows Michaelis–Menten kinetics, with a K$_m$ of 1.11 mM ± 0.20 mM and a v$_{max}$
*Figure 5. Continued on next page*

*Figure 5. Continued*

of 1226 ± 77 nmol · min⁻¹ · mg⁻¹. (**C**) At pH 8, $K_m$ drops to 0.84 ± 0.11 mM and $v_{max}$ increases to 2187 ± 88 nmol · min⁻¹ · mg⁻¹ (1.68 s⁻¹).

The following figure supplement is available for figure 5:

**Figure supplement 1**. Na⁺-dependent transport activity on a linear scale.

**Table 2.** Electron crystallographic data

| | pH 4, 0 mM NaCl |
|---|---|
| Unit cell dimensions | a = 81.5 Å, b = 103.3 Å, c = 200 Å, γ = 90° |
| Two-sided plane group | $p22_12_1$ |
| Number of images (tilt angles in brackets) | 15 (0°), 23 (20°), 44 (30°), 46 (45° and above) |
| In-plane resolution | 6 Å |
| Resolution in z direction[a] | 14 Å |
| Defocus range | 0,12–1,8 μm |
| Tilt range | 0–54° |
| Total number of observed reflections[b] | 47,064 |
| Observed unique reflections | 15,509 |
| Unique reflections in asymmetric unit | 2686 |
| Overall weighted phase residual[b] | 12.1° |
| Overall weighted R-factor[b] | 24.8% |

[a]calculated from the point spread function of the experimental data.
[b]calculated with program LATLINEK.
[a,b] Reflections with IQ ≤ 6 Å were included.

which is part of the 6-helix bundle, impairs the mobility of the bundle at elevated pH. This movement is an integral feature of the transport mechanism. For reliable functional measurements on flexible, conformationally active membrane transporters it is therefore advisable to use untagged proteins. ²²Na⁺ uptake by untagged MjNhaP1 reconstituted at a high lipid/protein ratio into proteoliposomes acidified by an ammonium gradient also avoids other potential problems associated with the limited pH range of fluorescent dyes, everted vesicles energized by a process that is itself pH-dependent (*Reenstra et al., 1980*), or with leaky proteoliposomes produced at low lipid/protein ratio (*Tsai et al., 2013*).

The bell-shaped pH profile of MjNhaP1 and PaNhaP (*Wöhlert et al., 2014*) confirms an earlier conclusion that the antiporter has to shut down at acidic or basic pH for physiological reasons (*Vinothkumar et al., 2005*). Our finding that proton-driven Na⁺ uptake drops at decreasing external pH (*Figure 5A*) is in good agreement with the inverse experiment, which showed that Na⁺ efflux in MjNhaP1 proteoliposomes increases under these conditions (*Calinescu et al., 2014*). In the Na⁺ uptake experiment, substrate ions bind less well from the outside at low external pH due to proton competition. Conversely, in the efflux experiment (*Calinescu et al., 2014*), a decrease in external pH has no effect on Na⁺ binding from the inside.

The turnover number of MjNhaP1 was derived from $v_{max}$ = 2187 ± 88 nmol · min⁻¹ · mg⁻¹ at pH 8 measured at 0°C. At any higher temperature, transport was too fast to be reliably recorded. The measured rate of 1.68 ions per second per protomer is more than 400 times higher than the transport rate of PaNhaP extrapolated to 0°C. The difference is most likely due to an extra acidic sidechain (Glu73) in the ion-binding site of PaNhaP, which has no equivalent in MjNhaP1. Removal of this sidechain in PaNhaP increases the transport rate, because the substrate ion is released more readily (*Wöhlert et al., 2014*). Although compared to EcNhaA (*Taglicht et al., 1991*), neither MjNhaP1 nor PaNhaP are particularly fast at ambient conditions, the activity of PaNhaP increases exponentially with temperature to an extrapolated turnover number of 5000 ions per second at 100°C (*Wöhlert et al., 2014*). Given its similarity to PaNhaP, MjNhaP1 will be very much faster at its physiological temperature of 85°C than at room temperature, and most likely also considerably faster than PaNhaP under physiological conditions.

## Transport mechanism

The x-ray structure of MjNhaP1 was determined at pH 8 in the presence of substrate ions, where the antiporter is highly active. The x-ray structures of PaNhaP were determined at pH 4 or pH 8 also in

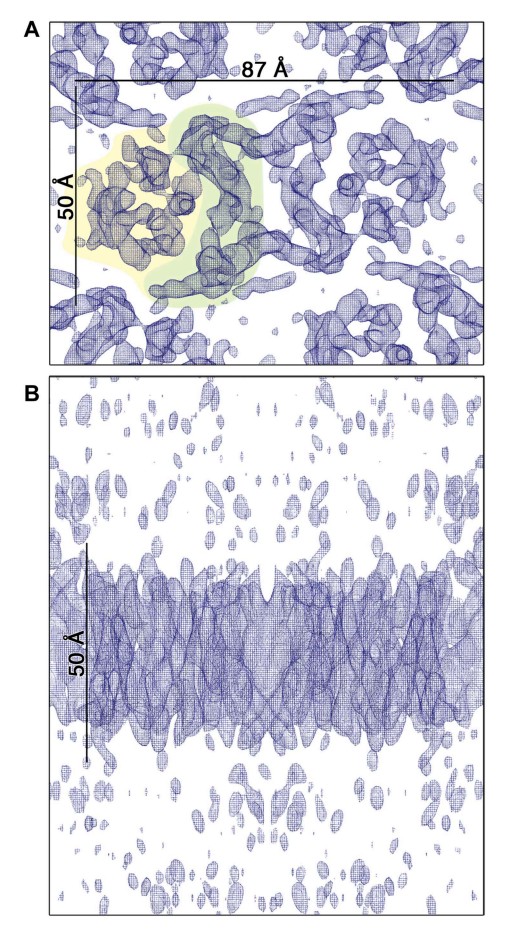

**Figure 6**. 3D EM map of MjNhaP1 at pH 4 in the absence of Na$^+$. The map was contoured at 1.7σ and a B-factor of −200 Å$^2$ was applied. (**A**) Cytoplasmic view of one dimer. The 6-helix bundle is shaded yellow and the dimer interface green. (**B**) The side view indicates low noise level outside the membrane.

The following figure supplement is available for figure 6:

**Figure supplement 1**. Electron crystallographic amplitudes and phases.

the presence of Na$^+$. Under both conditions PaNhaP is inactive (**Wöhlert et al., 2014**). The close resemblance of the MjNhaP1 and PaNhaP x-ray structures proves that there is no pH-induced conformational switch to regulate either antiporter. This is in excellent agreement with a recent study (**Paulino and Kühlbrandt, 2014**), which indicated that a change in pH in the absence of substrate ions has only minimal effects on the conformation of MjNhaP1, whereas Na$^+$-binding induces helix movements that are similar in the entire activity range and consistent with the changes in helix orientation described here (**Figure 10**). We propose that these considerations hold true for all CPA1 antiporters.

Like the mammalian NHE exchangers, the electroneutral Na$^+$/H$^+$ antiporters MjNhaP1 and PaNhaP are thought to work as Na$^+$-driven proton transporters. They maintain an intracellular neutral pH by utilizing the inward Na$^+$ gradient that is always present in their native saline habitat. In PaNhaP, Na$^+$ coordination stabilizes the inward-open state, whereas the apo or proton-bound state of MjNhaP1 adopts the outward-open conformation (**Paulino and Kühlbrandt, 2014**). Thus, their default resting state is inward-open with a sodium ion in the binding site. As suggested for EcNhaA (**Mager et al., 2011**) and recently shown for MjNhaP1 (**Calinescu et al., 2014**; **Paulino and Kühlbrandt, 2014**), protons and Na$^+$ compete for a single binding site in the protomer. When the intracellular pH drops, the binding site becomes protonated and the Na$^+$ ion is released into the cytoplasm. The conformational changes we observe are consistent with a rocking bundle mechanism, as proposed for other secondary transporters (**Forrest and Rudnick, 2009**). The rocking movement of the 6-helix bundle controls alternating access to the ion-binding site from the outside medium or from the cell interior (**Video 3**).

In summary, the simplest mechanism for MjNhaP1 and other CPA1 antiporters entails the following four steps (**Figure 11**). (i) In the default resting state, a proton from within the cell replaces the bound Na$^+$, which is released to the cytoplasm; (ii) upon protonation of Asp161, the antiporter switches to the outward-open state, making the ion-binding site accessible to extracellular Na$^+$; (iii) a Na$^+$ ion diffusing to the binding site from the extracellular medium displaces the proton at Asp161 or its equivalent; the deprotonated sidechain engages in Na$^+$ coordination; (iv) deprotonation and Na$^+$ binding triggers the switch from the outward-open back to the inward-open state. Na$^+$ is released into the cytoplasm, a proton binds, and the cycle repeats.

The most important remaining questions concern the exact molecular events during the transition from the inward-open to the outward-open state, and the underlying energetics. These questions are best addressed by molecular dynamics simulations. Given that its structures are now available in both the inward-open and the outward-open state and its transport cycle may take only 2 μs, MjNhaP1 would be an ideal target.

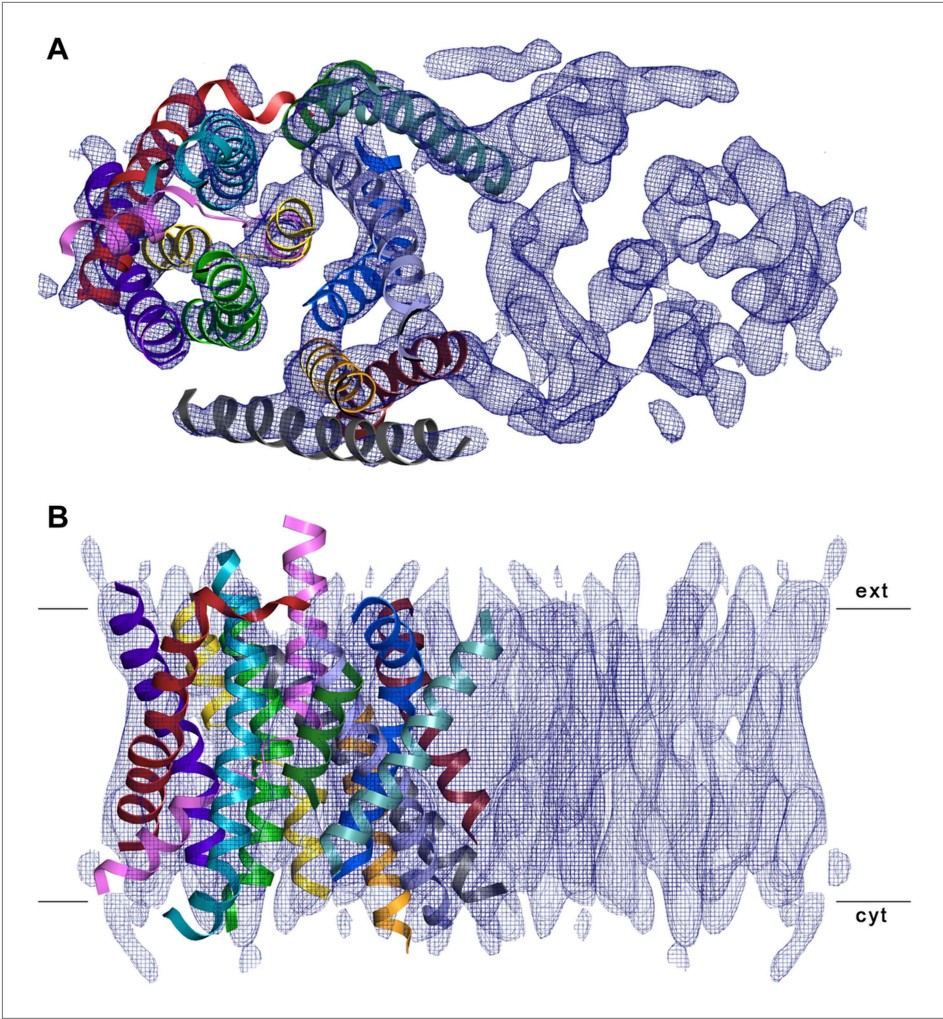

**Figure 7**. 3D EM structure of MjNhaP1 at pH 4 without sodium. Cytoplasmic (**A**) and side view (**B**) of 3D EM map with one MjNhaP1 protomer fitted. The map was sharpened with a B factor of −200 Å² and contoured at 1.7σ. Connecting helices and loops without EM density were omitted for clarity.

## Materials and methods

### Membrane topology

The relative orientation of the protein in the membrane was determined by GFP/PhoA fusion (*Daley et al., 2005*; *Drew et al., 2002*; *ter Horst and Lolkema, 2012*). For the GFP assay, MjNhaP1 or EcNhaA genes were cloned into the pWaldo plasmid carrying a C-terminal GFP-tag, using the restriction sites *XhoI* and *EcoRI* and the following primers:

MjNhaP_XhoI_s: 5′-CCGCCGCTCGAGATGGAACTGATGATGGCGATCG-3′
MjNha1_EcoRI_as: 5′-CGGCGGGAATTCATGGTGGCTTTCTTCTTTATATTTCG-3′
EcNhaA_XhoI_s: 5′-CCGCCGCTCGAGGTGAAACATCTGCATCGATTC-3′
EcNhaA_EcoRI_as: 5′-CGGCGGGAATTCAACTGATGGACGCAAACGAAC-3′

BL21(DE3) cells were transformed and grown at 37°C in 10 ml LB medium with 50 µg/ml kanamycin. Protein expression was induced at an $OD_{600}$ of ~0.4 by addition of 1 mM IPTG and the cells were harvested at an $OD_{600}$ of 0.5–0.6. The cell pellet was washed 50 mM Tris/HCl pH 8, 200 mM NaCl and 15 mM EDTA and resuspended in 300 µl of the same buffer. Samples were analyzed by in-gel fluorescence and Western blot analysis using α-His or α-GFP antibodies. 180 µl of whole cell suspension were transferred into 96-well Nunc plates and incubated for 1.5 hr. GFP fluorescence was measured at 512 nm with excitation at 485 nm. The mean fluorescence was calculated from three

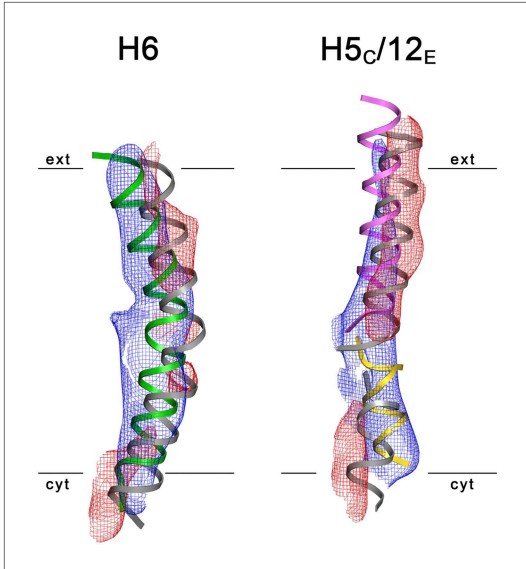

**Figure 8**. 3D Difference maps. Difference densities were calculated between the unsharpened experimental EM map (blue mesh) and the x-ray map truncated to 6 Å resolution. Difference maps are shown for H6 (green) and the $5_C/12_E$ pair of half helices (yellow/pink). For clarity, only negative difference densities are shown (red mesh). The x-ray structure of the inward-open state is grey. Maps were plotted at $2\sigma$.

independent measurements and normalized to respective $OD_{600}$. Untransformed BL21(DE3) cells were used as negative control and the EcNhaA constructs as positive controls. For the PhoA assay, MjNhaP1 was cloned into the pHA-1 plasmid carrying a C-terminal PhoA-tag, using the restriction sites *XhoI* and *BsiWI/Acc651* and the following primers:

MjNhaP1_XhoI_s: 5'-CCGCCGCTCGAGATGGAACTGATGATGGCGATCG-3'

MjNhaP1_BsiWI_as: 5'-GGCGGCGTACGGCATGGTGGCTTTCTTCTTTATATTT-3'

Plasmids carrying PhoA-fusions constructs of known topology (pos: YiaD; neg: YedZ) were kindly provided by Gunnar von Heijne and used as controls. The PhoA deficient *E. coli* strain CC118 was transformed and grown in 10 ml LB medium with 100 µg/ml ampicillin at 37°C. Protein expression was induced with 0.2% arabinose at $OD_{600}$ = 0.15, cultures were harvested after 2–2.5 hr and 1 mM iodoacetamide was added. The optical density for all cultures was measured and normalized. Cell pellets were washed and resuspended in 10 mM Tris/HCl pH 8 and 1 mM iodoacetamide. For the PhoA assay, 0.2 ml of the cell suspension were added to 0.8 ml 1 M Tris/HCl pH 8, 0.1 mM $ZnCl_2$. To permeabilize the cells, 50 µl 0.1% SDS and 50 µl chloroform were added to the reaction mixture. The cells were incubated 5 min at 37°C and 5 min on ice. The reaction was started by addition of 0.1 ml p-nitrophenyl phosphate. After incubation at 37°C for 90 min, the reaction was stopped by addition of 120 µl 1:5 0.5 M EDTA pH 8, 0.1 M $KH_2PO_4$. For each sample the absorption at 420 nm and 550 nm was recorded and the mean activity was obtained from four independent measurements. For Western blot analysis, 50 µl sample was centrifuged and cell pellet was resuspended in 20 µl SDS sample buffer. A PhoA antibody was used for fusion-protein detection.

## Cloning, expression and purification

For 3D crystallization and $^{22}Na^+$ uptake measurements MjNhaP1 was cloned into a pET-21a vector with a C-terminal Cysteine Protease Domain (CPD) fusion (*Shen et al., 2009*) that produces untagged protein. *E. coli* BL21-(DE3) cells were transformed with the resulting plasmid and MjNhaP1 was expressed at 37°C in ZYM-5052 autoinduction medium (*Studier, 2005*). Cultures were harvested and cells were broken with a microfluidizer (M-110L, Microfluidics Corp., Westwood, MA). Membranes were isolated by centrifugation at 100,000×$g$ at 4°C for 1 hr, resuspended in 50 mM Tris/HCl pH 7.5, 140 mM choline chloride, 250 mM sucrose and diluted 1:2 in 150 mM MOPS/KOH pH 7.0, 45% glycerol and ~2.0% Foscholine-12. After incubation at 4°C for 2 hr the solution was clarified by centrifugation at 125,000×$g$ for 1 hr. The supernatant was supplemented with 5 mM imidazole and 150 mM NaCl, incubated for 2 hr with TALON resin (Clontech, Mountain View, CA) at 4°C and loaded on a Biorad column. Unspecifically bound protein was eluted with 15 column volumes of 20 mM Bis-Tris pH 7.0, 300 mM NaCl, 10 mM imidazole, 0.1% Foscholine-12, 0.24% Cymal-5 and 20 column volumes of 20 mM Bis-Tris pH 6.5, 0.2% Cymal-5, 300 mM NaCl. MjNhaP1 was eluted from the column with 10 column volumes of 20 mM Bis-Tris pH 6.5, 300 mM NaCl, 0.2% cymal-5, 10 µM inositol-hexaphosphate, concentrated to 8 mg/ml using a concentrator with 50 kDa cut-off and dialysed against 25 mM Sodium-Acetate pH 4.0, 100 mM NaCl, 0.2% Cymal-5.

For 2D crystallization MjNhaP1 was expressed in the pET26b vector with a C-terminal hexa-histidine tag and purified by Ni-NTA affinity chromatography as described (*Paulino and Kühlbrandt, 2014*). To ensure Na$^+$-free conditions the column was washed with 10 column volumes of 15 mM Tris/HCl pH 7.5,

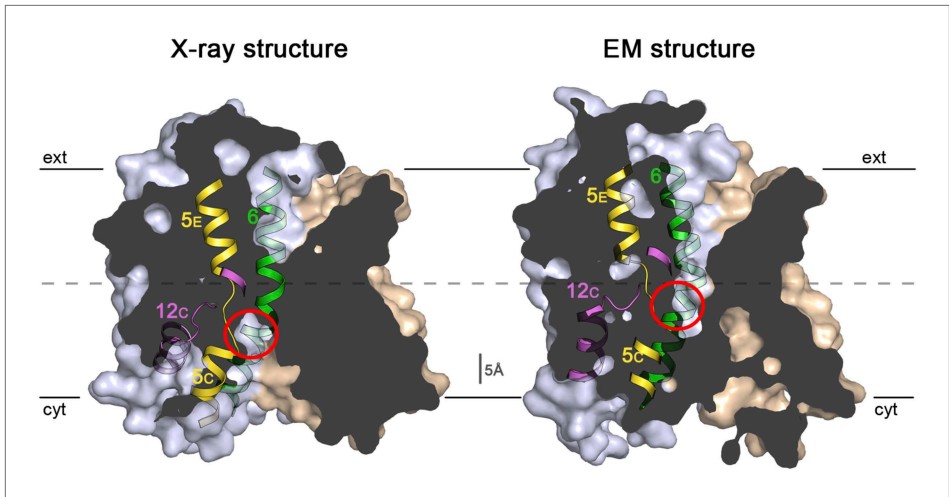

**Figure 9**. Sections through the MjNhaP1 x-ray and EM structures. Sections through the inward-open x-ray structure and the outward-open EM structure of MjNhaP1. In the x-ray structure (left) the ion-binding site (red circle) is accessible from the cytoplasm. In the EM structure (right), the ion-binding site has moved upwards by ~5 Å and is accessible through the extracellular funnel.

The following figure supplement is available for figure 9:

**Figure supplement 1**. Comparison of MjNhaP1 and TtNapA.

500 mM NaCl, 15 mM imidazole and 0.03% dodecyl maltoside (DDM), followed by 8 column volumes of sodium-free buffer (15 mM Tris/HCl pH 7.5, 200 mM KCl and 0.03% DDM). MjNhaP1 was eluted with 50 mM potassium acetate pH 4, 100 mM KCl, 5 mM MgCl$_2$ and 0.03% DDM, concentrated and stored at −80°C.

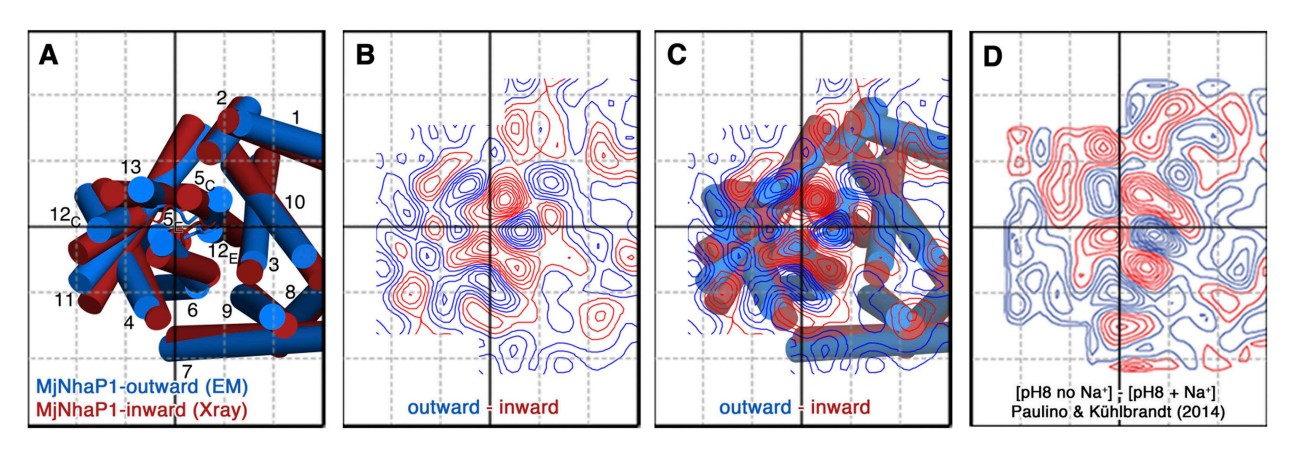

**Figure 10**. Projection difference maps. (**A**) Superposition of the three-dimensional MjNhaP1 outward-open structure at pH 4 without sodium (blue) and the inward-open structure at pH 8 with sodium (red). Helices are shown as cylinders as seen from the cytoplasmic side. (**B**) 6 Å projection difference map calculated between the structures shown in (**A**). Major difference peaks are observed in the 6-helix bundle, whereas difference peaks at the dimer interface are weak. (**C**) Superposition of the inward-open and outward-open MjNhaP1 structures on the projection difference map shown in (**B**). (**D**) 6 Å projection difference map between MjNhaP1 2D crystals at pH 8 with and without sodium (adapted from *Figure 5* in (*Paulino and Kühlbrandt, 2014*) top row, 500 mM NaCl brought to the same phase origin). The projection difference map calculated from the 3D structures closely resembles the projection map that shows sodium-induced changes in *Paulino and Kühlbrandt (2014)*. The conformational change from the inward-open to outward-open state of MjNhaP1 is therefore induced by sodium ions.

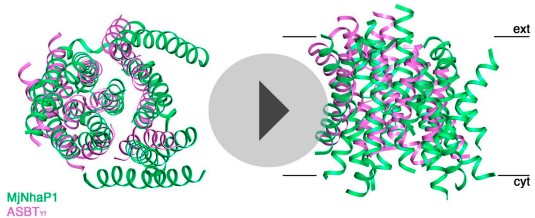

**Video 2**. Conformational changes in MjNhaP1 and ASBT. Morphing the transition from the outward-open to the inward-open states in MjNhaP1 (green) and ASBT$_{Yf}$ (*Zhou et al., 2014*) (purple) reveals a very similar rigid-body movement of the 6-helix bundle relative to the dimer interface in both proteins. Cytoplasmic view (left) and side view (right). Structures were superimposed on the dimer interface and intermediate states were calculated using the program LSQMAN.

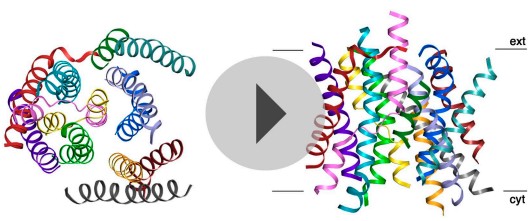

**Video 3**. Transport cycle of MjNhaP1. Substrate-induced conformational changes in MjNhaP1 from the outward-open sodium-free state at pH 4 to the inward-open state at pH 8 in the presence of sodium. The 6-helix bundle rotates by ~7° with respect to the dimer interface, and the ion-binding site moves by ~5 Å, resulting in alternating access to the ion-binding site from the cytoplasm or from the extracellular side. Cytoplasmic view (left) and side view (right). Structures were superimposed on the dimer interface and intermediate states were calculated using the program LSQMAN.

## 3D crystallization, x-ray crystallography, data processing and structure determination

Prior to 3D crystallization, MjNhaP1 was incubated at 85°C for 15 min and centrifuged for 1 hr at 125,000×*g*. The supernatant was passed through a 0.1 µm filter (Ultrafree-MC, Millipore, Billerica, MA), supplemented with 5 mM $K_2Pt(CN)_4$ and mixed 1:1 with 100 mM Tris/Cl pH 8.2, 24% PEG 1000. MjNhaP1 crystals grew in hanging drops within 10–14 days to a maximal size of 350 µm and were vitrified directly in liquid nitrogen. Data were collected at the ESRF beamline id23.1, processed with XDS (*Kabsch, 1993*) and scaled with AIMLESS in the CCP4 package (*Collaborative Computational Project 4, 1994*). Resolution cut-offs were set on the basis of cross correlation between half datasets, completeness and I/σ(I)-values in high resolution shells (*Karplus and Diederichs, 2012*). COOT (*Emsley and Cowtan, 2004*) was used for model building and the PHENIX package (*Adams et al., 2010*) for refinement. Phases were obtained by molecular replacement with PHASER (*McCoy, 2007*) using a polyalanine dimer model of the PaNhaP x-ray structure (4cz8) as search template. After density modification with Parrot (*Zhang et al., 1997*) the protein backbone was rebuild into the density-modified map and a run of molecular replacement was started with the new template. This process was repeated several times. After correcting the backbone geometry, side chains were fitted to the electron density, starting with the residues that are conserved between PaNhaP and MjNhaP1 in several rounds of iterative model building and refinement using phenix. refine (*Adams et al., 2010*).

## 2D crystallization, electron crystallography, image processing and model building

2D crystals of MjNhaP1 were grown with *E. coli* polar lipids (Avanti Polar Lipids, Inc., Alabaster, AL) at a final protein concentration of 1 mg/ml, a final decyl maltoside concentration of 0.15% and a lipid-to-protein ratio (LPR) of 0.5 (*Paulino and Kühlbrandt, 2014*). 2D crystals were grown at 37°C in sodium-free 25 mM K⁺ acetate pH 4, 200 mM KCl, 5% glycerol and 5% 2-4-methylpentanediol. EM grids were prepared by the back-injection method (*Wang and Kühlbrandt, 1991*) in 4% trehalose and rapidly frozen in liquid nitrogen. Images were recorded with an electron dose of 20–30 e⁻/Å² on Kodak SO-163 film with a JEOL 3000 SFF electron microscope at a nominal temperature of 4K, an acceleration voltage of 300 kV, a magnification of 53,000 at a defocus range of 0.1–1.8 µm in spot scan mode. Images of tilted crystals were recorded with fixed tilt angle cryo holders. Lattice images were screened by optical diffraction, and well-ordered areas of 4k × 4k or 6k × 6k pixels were digitized at 7 µm step size on a Zeiss SCAI scanner. Images were processed by the 2dx software package (*Gipson et al., 2007*).

A total of 128 image areas (15 at 0°, 23 at 20°, 44 at 30° and 46 at 45° or above) were used for 3D reconstruction. Data quality was improved by synthetic unbending (*Arheit et al., 2013*). To compensate for the resolution-dependent degradation of image amplitudes a negative temperature factor of

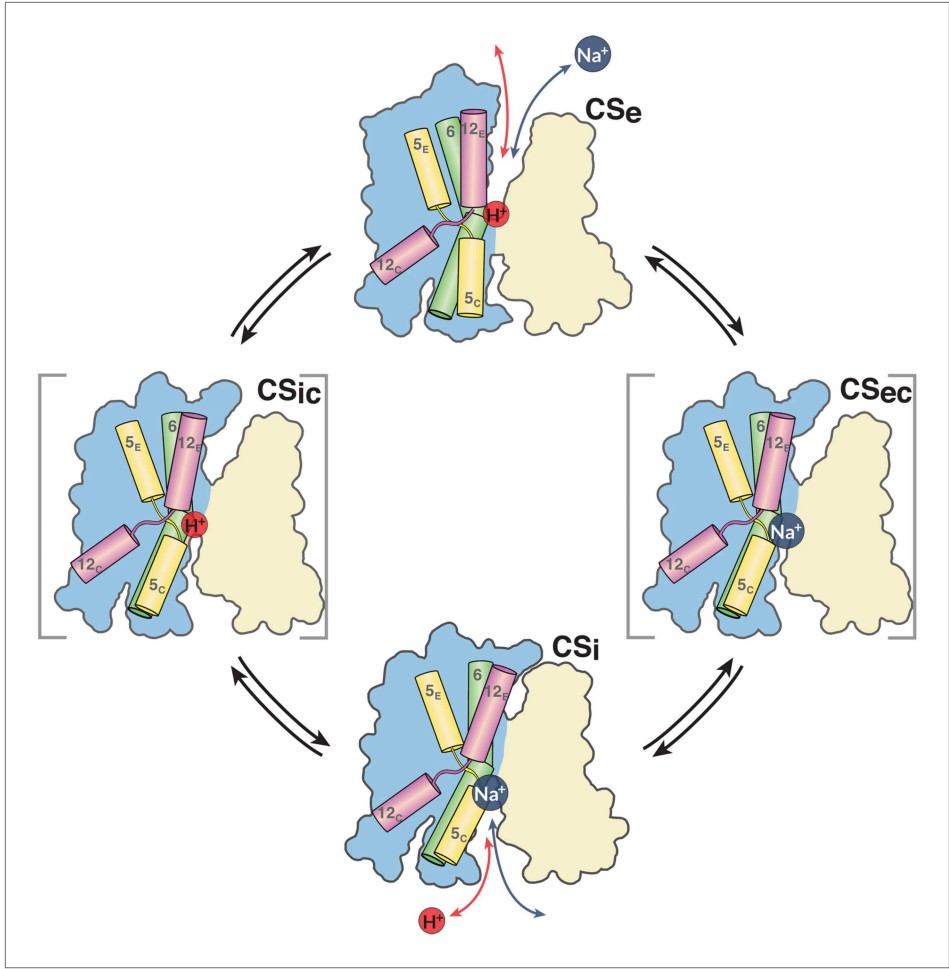

**Figure 11**. The MjNhaP1 transport cycle. In the outward-open state (Ce), Na$^+$ from the exterior medium gains access to the ion-binding site through the open extracellular funnel, where it replaces a bound proton. Na$^+$ binding triggers the transition from the outward-open to the inward-open, substrate-bound conformation (C$_{Si}$) via a substrate-occluded state (CS$_{ec}$). In the inward-open state, the cytoplasmic funnel widens and the extracellular funnel closes. A cytoplasmic proton releases the bound Na$^+$ to the cell interior. Na$^+$ release triggers the conformational change to the outward-open state via an occluded inward-open proton-bound conformation (CS$_{ic}$), and the cycle repeats. H5, 6 and 12 are color-coded. The 6-helix bundle (blue) performs a ~7° rocking motion around an axis parallel to the dimer interface (light brown), which remains fixed in the membrane.

B = −200 Å$^2$ was used. A molecular model was build by fitting the MjNhaP1 x-ray structure into the 3D EM density in COOT (*Emsley and Cowtan, 2004*). 3D difference maps were calculated with scripts from the CCP4 (*Winn et al., 2011*) and the PHENIX (*Adams et al., 2010*) software packages (as indicated below). The EM and x-ray density maps were expanded, scaled and the resolution cut to 6 Å (sftools). Maps were superimposed in PHENIX (*Adams et al., 2010*) and placed into a cell with identical units (mapmask, maprot). The density of one protomer was masked with the help of the pdb models (pdbset, ncsmask, maprot, mapmask), and both maps were subtracted from one another (overlapmap).

## Figures and movies

Figures and movies were prepared with PyMOL (*DeLano and Lam, 2005*). Superimpositions were performed using Secondary Structure Superimposition within COOT (*Emsley and Cowtan, 2004*; *Krissinel and Henrick, 2004*). For morphing LSQMAN (*Kleywegt, 1996*) from the Uppsala Software Factory was used to generate a series of intermediates between structures. Sequence alignments were performed using ClustalX (*Larkin et al., 2007*) and adjusted in JalView (*Waterhouse et al., 2009*).

The potential surface was calculated with pdb2pqr (*Dolinsky et al., 2007*) and APBS (*Baker et al., 2001*). Analysis of transport pathways, channels and cavities was performed with Hollow (*Ho and Gruswitz, 2008*) and visualized within PyMOL.

## Proteoliposome activity assays

For sodium efflux under symmetrical pH, *E. coli* polar lipids were dried under nitrogen and resuspended in 10 mM choline citrate/Tris/glycine pH 6–9, 5 mM KCl, 200 mM sodium chloride, 10 mM β-mercaptoethanol. Liposomes were preformed using polycarbonate filters with a pore size of 400 nm and destabilized by addition of octyl glucoside to a final concentration of 1%. MjNhaP1 was added at an LPR of 200:1 and the suspension was incubated for 1 hr. Detergent was removed by dialysis (14 kDa cut-off) overnight against detergent-free reconstitution buffer. To ensure complete detergent removal, 1 g of Biobeads were added to the dialysis buffer per 4 ml of lipids. Proteoliposomes were centrifuged at 300,000×$g$ for 20 min and washed once with reconstitution buffer. Liposomes were centrifuged again and resuspended at a lipid concentration of 60 mg/ml in reconstitution buffer. 2 μl of proteoliposomes were diluted in 2 ml of reaction buffer (10 mM choline citrate/Tris/glycine at the same pH, 5 mM KCl, 2 μM acridine orange) to start the reaction. Antiport activity of MjNhaP1 establishes a ΔpH across the membrane, observed as acridine orange quenching. As a control, $(NH_4)_2SO_4$ was added to a final concentration of 25 mM at the end of the reaction to dissipate the pH gradient. Measurements were performed at 25°C and acridine orange fluorescence was monitored at 530 nm (excitation: 495 nm) in a Hitachi fluorimeter.

For $^{22}$Na uptake measurements, reconstitution was performed as described for sodium efflux under symmetrical pH conditions, with the following changes. Lipids were resuspended in 20 mM choline citrate/Tris/glycine pH 4–10, 10 mM $(NH_4)_2SO_4$, 10 mM β-mercaptoethanol and the LPR was 400:1. Transport was initiated by diluting 2 μl of proteoliposomes into 200 μl of reaction buffer (20 mM choline citrate/Tris/glycine at the same pH, 10 mM choline chloride, 2 mM $MgSO_4$) containing NaCl at final concentrations between 25 μM and 6.5 mM and 1 μCi/ml $^{22}$Na. Dilution of proteoliposomes in $(NH_4)_2SO_4$-free reaction buffer results in $NH_3$ efflux, acidifying the interior of the liposomes (*Dibrov and Taglicht, 1993*). Transport was stopped by filtering the proteoliposomes on 0.2 μm nitrocellulose filters and washing with 3 ml ice-cold $^{22}$Na-free reaction buffer. Before counting, filters were transferred to counting tubes and 4 ml scintillation cocktail (Rotiszint, Roth, Germany) was added. All measurements were performed on ice and repeated at least three times.

## Mutagenesis studies

The activities of N-terminal truncated MjNhaP1 constructs and the Asn160 mutant were determined under asymmetrical pH by fluorescence in everted vesicles or proteoliposomes (*Goswami et al., 2011*). MjNhaP1 constructs were cloned into the pTrcHis2-Topo plasmid, carrying a C-terminal Myc-His tag, via *NcoI* and *EcoRI* restriction sites and the following primers:

MjNhaP_6s_NcoI: 5′-CCGCCGCCATGGCCCTTGCTATTGGTTACCTTGGATAGC-3′
MjNhaP_10s_NcoI: 5′-CCGCCGCCATGGCCCTTGGATTAGCTTTAGTTCTTGGTTC-3′
MjNhaP_16s_NcoI: 5′-CCGCCGCCATGGCCCTTCTTGGTTCGTTAGTGGCAAAAATTG-3′
MjNhaP_426as_EcoRI: 5′-CAAAGTATAAAGAAGAATCCCACCATAAGGGCGAATTCGCCGCC-3′

Point mutations were generated using the QuickChange II Site-Directed Mutagenesis Kit (Agilent Technologies) and the following primers:

MjNhaP1_N160A_s: 5′-GTTAGAGGCGGAGAGTATCTTTGCCGACCCATTGGGAATAGTTTC-3′
MjNhaP1_N160A_as: 5′-GAAACTATTCCCAATGGGTCGGCAAAGATACTCTCCGCCTCTAAC-3′
MjNhaP1_N160D_s: 5′-GTTAGAGGCGGAGAGTATCTTTGACGACCCATTGGGAATAGTTTC-3′
MjNhaP1_N160D_as: 5′-GAAACTATTCCCAATGGGTCGTCAAAGATACTCTCCGCCTCTAAC-3′

## Author information

Coordinates and structure factors for the pH 8 X-ray structure and the pH 4 EM structure were deposited in the PDB with the accession code 4czb and 4d0a, respectively. The 3D EM map was deposited in the EM data bank with the accession code EMD-2636.

## Acknowledgements

We thank Deryck Mills for EM maintenance and management; Sabine Häder for technical assistance. Gerhard Hummer, Klaus Fendler and Christine Ziegler for discussions, Marcel Arheit and Henning Stahlberg (C-CINA, Basel) for software support; Gunnar von Heijne for GFP/PhoA constructs and

CC118 cells. Crystals were screened at the Max Planck beamline PXII of the Swiss Light Source (SLS) and x-ray data were collected at ESRF beamline id23.1. This work was funded by the Max Planck Society; the Frankfurt Cluster of Excellence Macromolecular Complexes; the Frankfurt International Max Planck Research School; and SFB 807 'Transport and communication across biological membranes'. CP acknowledges support from the FCT PhD fellowship (SFRH/BD/62643/2009, Portugal).

## Additional information

### Competing interests

WK: Reviewing editor, eLife. The other authors declare that no competing interests exist.

### Funding

| Funder | Grant reference number | Author |
|---|---|---|
| Max-Planck-Gesellschaft (Max Planck Society) | | Cristina Paulino, David Wöhlert, Ekaterina Kapotova, Özkan Yildiz, Werner Kühlbrandt |
| Deutsche Forschungsgemeinschaft (DFG) | Cluster of Excellence, Macromolecular Complexes | Werner Kühlbrandt |
| Max-Planck-Gesellschaft (Max Planck Society) | International Max Planck Research School, Frankfurt | Cristina Paulino, David Wöhlert, Ekaterina Kapotova, Werner Kühlbrandt |
| Deutsche Forschungsgemeinschaft (DFG) | Transport and communication across biological membranes SFB807 | Cristina Paulino, David Wöhlert, Werner Kühlbrandt |
| Fundação para a Ciência e a Tecnologia (Foundation for Science and Technology) | SFRH/BD/62643/2009 | Cristina Paulino |

The funders had no role in study design, data collection and interpretation, or the decision to submit the work for publication.

### Author contributions

CP, Purified and crystallized protein in 2D, Collected EM data, Analyzed and interpreted EM data, Performed functional analyses, Acquisition of data, Analysis and interpretation of data, Drafting or revising the article; DW, Collected x-ray data, Analyzed data and solved the structure, Performed functional analyses, Acquisition of data, Analysis and interpretation of data, Drafting or revising the article; EK, Purified the protein, Optimized the 3D crystals; ÖY, Collected and analyzed x-ray data, Conception and design, Acquisition of data, Analysis and interpretation of data, Drafting or revising the article; WK, Analyzed and interpreted EM data, Conception and design, Analysis and interpretation of data, Drafting or revising the article

## Additional files

### Major datasets

The following datasets were generated:

| Author(s) | Year | Dataset title | Dataset ID and/or URL | Database, license, and accessibility information |
|---|---|---|---|---|
| Woehlert D, Paulino C, Kapotova E, Kuhlbrandt W, Yildiz O | 2014 | Structure of the sodium proton antiporter MjNhaP1 from Methanocaldococcus jannaschii at pH 8 | http://www.rcsb.org/pdb/search/structidSearch.do?structureId=4czb | Publicly available at RCSB Protein Data Bank. |
| Paulino C, Woehlert D, Yildiz O, Kuhlbrandt W | 2014 | 3D EM map of the sodium proton antiporter MjNhaP1 from Methanocaldococcus jannaschii | http://www.rcsb.org/pdb/search/structidSearch.do?structureId=4d0a | Publicly available at RCSB Protein Data Bank. |

| Paulino C, Woehlert D, Yildiz O, Kuhlbrandt W | 2014 | 3D EM map of the sodium/proton antiporter MjNhaP1 | http://www.ebi.ac.uk/pdbe/entry/EMD-2636 | Publicly available at the Electron Microscopy Data Bank. |

The following previously published datasets were used:

| Author(s) | Year | Dataset title | Dataset ID and/or URL | Database, license, and accessibility information |
|---|---|---|---|---|
| Hunte C, Screpanti E, Venturi M, Rimon A, Padan E, Michel H | 2005 | Structure of a Na+/H+ antiporter and insights into mechanism of action and regulation by pH | http://www.rcsb.org/pdb/explore/explore.do?structureId=1zcd | Publicly available at RCSB Protein Data Bank. |
| Drew D, Beckstein O, Lee C, Yashiro S, Sansom MSP, Iwata S, Cameron AD | N/A | A Membrane Embedded Salt-Bridge in the Sodium Proton Antiporter Nhaa | http://www.rcsb.org/pdb/explore/explore.do?structureId=4au5 | Publicly available at RCSB Protein Data Bank. |

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
