## [Decision Letter]

Thank you for sending your work entitled “Structure and transport mechanism of
the sodium/proton antiporter MjNhaP1” for consideration at
*eLife*. Your article has been evaluated by Michael Marletta (Senior
editor), Richard Aldrich (Reviewing editor), and 2 reviewers.

The Reviewing editor and the reviewers discussed their comments before we reached this
decision, and the Reviewing editor has assembled the following comments.

This paper, like its companion, looks at structures of Na/H exchangers homologous to the
NHE family in mammals. Here, the authors present an x-ray structure of MjNhaP1 (at pH
8.0), a close homolog of the PaNhaP presented in the companion paper. Because they could
not obtain crystals of MjNhaP1 at low pH they present a 3D EM map of the protein at low
pH and suggest, based on differences between this map and the x-ray model, that
conformational changes demonstrate the EM map to be in an outward facing state.
Unfortunately, the methods used to compare EM map and x-ray model are poorly justified
and the results are not convincing. In particular many of the differences between the
maps are slight vertical displacements, in the direction where the resolution of the EM
map is quite limited (Figure 8), as can easily be
seen in the EM density (Figure 6). The
accompanying biochemical experiments are nicely performed but the overall significance
of the high pH structure alone is not compelling.

Additional issues:

1) The structure of MjNhaP1 at pH 8, solved by molecular replacement with PaNhaP
(reported in the accompanying paper), is also “inward open” but has an
extracellular funnel that is surprisingly wide and deep. It was not clear whether this
funnel is seen in the putative outward facing conformation derived from the 3D EM
structure, and the authors do not discuss the significance of its dimensions. Was there
also a narrow cytoplasmic channel analogous to the one seen in PaNhaP?

2) The authors extrapolate their measured transport rate (1.68 ions per second) to a
number of 500,000 ions per second at 85 degrees. We strongly doubt that this is an
accurate number as this is essentially the turnover rate of an ion channel and is much
faster than any transporter we know. Many other factors are likely to come into play to
limit the turnover of the protein at these high temperatures and this misleading
speculation should be removed, especially from the Abstract.

3) The kinetic data presented in Figure 5
does not seem to be approaching saturation; it is therefore not possible to rely on the
fits of these data to obtain kinetic parameters.

4) The acridine orange assay used in both papers to measure proton flux is an excellent
assay for qualitative assessment of proton flux. However, the actual mechanism of
acridine orange is unknown in detail and it is impossible to quantitatively measure pH
change with this assay. Therefore the relative rates as a function of pH in Figure 5 are unreliable and should be omitted. Na22
flux could be used to measure these rates if desired, or a more quantitative pH probe,
like pyranene.

5) The close similarity of the active MjNhaP and inactive PaNhaP structures at pH 8
argue against large pH induced conformational changes that drive transporter activation.
This is a useful finding of this study and an excellent counter to the criticism that
both structures reported in accompanying paper of Wohlert et al. represent the inactive
transporter. In this respect, it would have been helpful to see the MjNhaP and PaNhaP
structures presented side by side in the same paper.

6) There are some interesting differences between MjNhaP and PaNhaP including an absence
of pH-dependent cooperativity that is attributed to the absence of a titratable
histidine in H10 and the stabilizing effect of ionic interactions at the dimer
interface. It is noted that the replacement of this His with Cys crosslinks the dimer
interface in PaNhaP, with corresponding loss of function, but it was not shown that
formation of the disulfide bond is pH dependent. Therefore, the attribution of
non-cooperativity in MjNhaP to the absence of this His, or to stable salt bridges, is
speculative and could have been tested more thoroughly.

7) The protein was crystallized in the presence of sodium, but unfortunately a
corresponding density for substrate is not apparent. Functional studies by mutagenesis
are minimal and disappointingly restricted to Asn160 that is part of the conserved ND
motif. Replacement with the equivalent Asp in the DD motif of CPA2 transporters or with
Ala gave convincing, but unsurprising results given findings in other Na+/H+
antiporters. The paper would have benefitted from more thoughtful mutagenesis
experiments that bolstered the relatively weak structural advances made.

---

## [Author Response]

*This paper, like its companion, looks at structures of Na/H exchangers
homologous to the NHE family in mammals. Here, the authors present an x-ray structure
of MjNhaP1 (at pH 8.0), a close homolog of the PaNhaP presented in the companion
paper. Because they could not obtain crystals of MjNhaP1 at low pH they present a 3D
EM map of the protein at low pH and suggest, based on differences between this map
and the x-ray model, that conformational changes demonstrate the EM map to be in an
outward facing state. Unfortunately, the methods used to compare EM map and x-ray
model are poorly justified and the results are not convincing*.

We not only compare the models of the EM and x-ray structures. We have actually
calculated difference maps between the electron density obtained by x-ray
crystallography of 3D crystals and the experimental 3D map obtained by cryo-EM of 2D
crystals (Figure 8). This is not trivial and as
far as we are aware it has not been done before. The maps indicate clearly that the
differences between the inward-open and outward-open state we observe by this unique
combination of experimental techniques are real and significant.

We did obtain 3D crystals of MjNhaP1 at low pH and low Na+ concentration, but they
do not diffract well enough for x-ray crystallography. This is why we used electron
crystallography to obtain the outward-open structure, as explained in the
manuscript.

*In particular many of the differences between the maps are slight vertical
displacements, in the direction where the resolution of the EM map is quite limited
(*Figure 8*),
as can easily be seen in the EM density (*Figure 6*). The accompanying biochemical
experiments are nicely performed but the overall significance of the high pH
structure alone is not compelling*.

We have now added a projection difference map between the inward-open x-ray structure
and the outward-open EM structure (Figure 8–figure supplement 1B, C). This map
provides incontrovertible evidence that the helix movements in the six-helix bundle are
mainly lateral, not vertical. Since several of the helices in MjNhaP1 are tilted, they
can be fitted unambiguously to the cryo-EM map not only in the horizontal but also in
the vertical direction, as shown clearly in Figure 7. This criticism raised by the reviewers is therefore invalid.

Furthermore, we compare this difference map (Figure 8–figure supplement 1B, C) to
the projection difference map in an earlier paper on substrate-ion induced
conformational changes in MjNhaP1 (Paulino & Kühlbrandt,
*eLife* 2014). Figure 8–figure supplement 1D in our revised
manuscript is an adaptation of Figure 5 in that
paper (top row, 500 mM NaCl, brought to the same phase origin for consistency). The
virtually identical positions and intensities of difference peaks in the two maps
indicate clearly that the difference between no salt and 500 mM NaCl at pH8 (or at pH4,
for that matter, see bottom row in Figure 5 of
Paulino & Kühlbrandt, *eLife* 2014) is the same as between the
inward-open and outward-open conformation of MjNhaP1. This allows us to relate the
conformational changes we observed in response to ionic conditions in the earlier paper
to the 3D structures of the inward-open or outward-open state presented in the present
manuscript. Note that this was not possible before, because the earlier paper (Paulino
& Kühlbrandt, *eLife* 2014) did not contain any 3D information.
The two studies are fully consistent with one another, and with the results of the
PaNhaP manuscript.

The observed helix tilts in the conformational change from the inward-open to the
outward-open state are of the order of ∼7°. This range is typical for state
changes in secondary transporters such as BetP, LeuT, Mhp1, ASBT. In particular the
conformational changes in MjNhaP1 and the structurally related bile acid symporter ASBT,
for which high-resolution x-ray structures of both the inward-open and the outward-open
state are available, are strikingly similar, as shown in Video 3. The conformational changes we see in
MjNhaP1 are thus exactly in line with those in other, well-characterized secondary
transporters.

*Additional issues*:

1) The structure of MjNhaP1 at pH 8, solved by molecular replacement with PaNhaP
(reported in the accompanying paper), is also “inward open” but has an
extracellular funnel that is surprisingly wide and deep. It was not clear whether
this funnel is seen in the putative outward facing conformation derived from the 3D
EM structure, and the authors do not discuss the significance of its dimensions. Was
there also a narrow cytoplasmic channel analogous to the one seen in
PaNhaP?

The substrate-binding site in the x-ray structure of MjNhaP1 at pH8 is accessible from
the cytoplasm but not from the extracellular side (Figure 9, Figure 2—figure supplement 1). Therefore this structure represents the inward-open state of MjNhaP1. In
the inward-open state, the extracellular funnel in MjNhaP1 is both deeper and wider than
that of PaNhaP, but it does not extend to the ion-binding site, as shown in Figure 9. In the outward-open EM structure the
extracellular funnel deepens and widens further, making the substrate-binding site
accessible from the extracellular side. In this state, access from the cytoplasm is
blocked by the cytoplasmic halves of helices 5 and 6. Hence the EM structure obtained at
pH4 in absence of sodium is in the outward-open state. This is clearly shown in Figure 9. The narrow cytoplasmic channel seen in
PaNhaP is not present in MjNhaP1, as shown in Figure 2—figure supplement 1.

*2) The authors extrapolate their measured transport rate (1.68 ions per second)
to a number of 500,000 ions per second at 85 degrees. We strongly doubt that this is
an accurate number as this is essentially the turnover rate of an ion channel and is
much faster than any transporter we know. Many other factors are likely to come into
play to limit the turnover of the protein at these high temperatures and this
misleading speculation should be removed, especially from the Abstract*.

The extrapolation is indeed likely to be inaccurate, as noted by the reviewers. However,
the extrapolated transport rate of MjNhaP1 is still far below the diffusion rate of ion
channels, which is at least two orders of magnitudes higher (>107 ions per second).
Compared with other antiporters at physiological temperatures, MjNhaP1 would be 100
times faster than PaNhaP (extrapolated to ∼5000 ions per second at 100°C, as
noted in the accompanying manuscript), but only 35 times faster than the widely accepted
transport rate of EcNhaA (extrapolated to 14,000 ions per second at 37°C, the
physiological temperature of E.coli). We do not see a problem with these high turnover
numbers, which seem feasible at high temperature, especially since the conformational
changes are small. We do agree with the reviewers however that the transport rate may be
limited by other factors, such as the surrounding lipids, which in our experiments are
different from the host lipids. As requested, we have taken the extrapolated turnover
rate out of the Abstract.

*3) The kinetic data presented in*
Figure 5
*does not seem to be approaching saturation; it is therefore not possible to rely
on the fits of these data to obtain kinetic parameters*.

In the revised Figure 5, panels B and C (same as
Figure 5 in the original manuscript)
show substantially smaller activity increases for the three data points at higher salt
concentration than in the mid-range of the curves. As a result the extrapolated
non-linear fit flattens towards the end of the curve, indicating saturation. This is
also reflected by the apparent half-saturation values that are in the middle of the
measured concentration range. That our measurements approach saturation conditions is
even clearer when the curves are plotted on a linear scale (Figure 5—figure supplement 1). The linear plots indicate
near saturation from around 5 mM NaCl.

*4) The acridine orange assay used in both papers to measure proton flux is an
excellent assay for qualitative assessment of proton flux. However, the actual
mechanism of acridine orange is unknown in detail and it is impossible to
quantitatively measure pH change with this assay. Therefore the relative rates as a
function of pH in*
Figure 5
*are unreliable and should be omitted. Na22 flux could be used to measure these
rates if desired, or a more quantitative pH probe, like pyranene*.

The transport rates as a function of pH in both papers were in fact measured by
22Na+ uptake. This was clearly stated in the original MjNhaP1 manuscript and in
Figure legend 5 and in 7 different places in the PaNhaP manuscript, which must have
escaped the reviewers’ attention. To avoid any possible confusion, we have
omitted the former Figure 5 from the revised
manuscript, which did show acridine orange measurements, but had been included only to
demonstrate transport under symmetrical pH conditions.

*5) The close similarity of the active MjNhaP and inactive PaNhaP structures at
pH 8 argue against large pH induced conformational changes that drive transporter
activation. This is a useful finding of this study and an excellent counter to the
criticism that both structures reported in accompanying paper of Wohlert et al.
represent the inactive transporter. In this respect, it would have been helpful to
see the MjNhaP and PaNhaP structures presented side by side in the same
paper*.

The new Video 1 in the revised manuscript
shows the differences between the MjNhaP1 and PaNhaP protomer in a 360°
rotation.

*6) There are some interesting differences between MjNhaP and PaNhaP including an
absence of pH-dependent cooperativity that is attributed to the absence of a
titratable histidine in H10 and the stabilizing effect of ionic interactions at the
dimer interface. It is noted that the replacement of this His with Cys crosslinks the
dimer interface in PaNhaP, with corresponding loss of function, but it was not shown
that formation of the disulfide bond is pH dependent. Therefore, the attribution of
non-cooperativity in MjNhaP to the absence of this His, or to stable salt bridges, is
speculative and could have been tested more thoroughly*.

This passage has been re-written in the revised manuscript, taking the reviewers’
comment into account. The original manuscript stated that His292 affects transport
activity in PaNhaP and that this residue is absent in MjNhaP1. However we did not
attribute the lack of cooperativity in MjNhaP1 to a single residue.

*7) The protein was crystallized in the presence of sodium, but unfortunately a
corresponding density for substrate is not apparent. Functional studies by
mutagenesis are minimal and disappointingly restricted to Asn160 that is part of the
conserved ND motif. Replacement with the equivalent Asp in the DD motif of CPA2
transporters or with Ala gave convincing, but unsurprising results given findings in
other Na+/H+ antiporters. The paper would have benefitted from more
thoughtful mutagenesis experiments that bolstered the relatively weak structural
advances made*.

The structural advances are not weak, as above. The focus of this manuscript is on the
observed conformational changes between the inward and outward-open conformation and new
insights into the transport mechanism. More mutagenesis studies are in progress but go
well beyond the scope of the present manuscript.